# The Potential of Non-Invasive Biomarkers for Early Diagnosis of Asymptomatic Patients with Endometriosis

**DOI:** 10.3390/jcm10132762

**Published:** 2021-06-23

**Authors:** Żaneta Kimber-Trojnar, Aleksandra Pilszyk, Magdalena Niebrzydowska, Zuzanna Pilszyk, Monika Ruszała, Bożena Leszczyńska-Gorzelak

**Affiliations:** 1Department of Obstetrics and Perinatology, Medical University of Lublin, 20-090 Lublin, Poland; apilszyk@gmail.com (A.P.); mniebrzydowska7@gmail.com (M.N.); monika.ruszala@wp.pl (M.R.); b.leszczynska@umlub.pl (B.L.-G.); 2Scientific Association at the 2nd Clinic of Gynecology and Obstetrics, Wroclaw Medical University, 50-367 Wrocław, Poland; z.pilszyk@gmail.com

**Keywords:** endometriosis, biomarker, diagnostic markers, CA-125, urocortin, activin A, follistatin, microRNA, urinary biomarkers

## Abstract

Endometriosis is a disease that affects women of reproductive age and has a significantly negative impact on their well-being. The main symptoms are dysmenorrhoea, chronic pelvic pain and infertility. In many patients the diagnostic process is very long and can take up to 8–12 years. Laparoscopy, an invasive method, is still necessary to confirm the diagnosis. Therefore, development of more effective diagnostic markers appears to be of the utmost importance for early diagnosis of endometriosis and provision of appropriate treatment. From a clinical point of view, detection of early-stage endometriosis in asymptomatic patients is an ideal situation since early diagnosis of endometriosis may delay the onset of symptoms as well as prevent progression and complications. In the meantime, Cancer Antigen 125 (CA-125) is still the most frequently studied and used marker. Other glycoproteins, growth factors and immune markers seem to play an important role. However, the search for an ideal endometriosis marker is still underway. Further studies into the pathogenesis of endometriosis will help to identify biomarkers or sets of biomarkers with the potential to improve and speed up the diagnostic process in a non-invasive way.

## 1. Introduction

Endometriosis is a progressive disease with features of chronic inflammation. It is still uncertain whether an inflammatory process is the cause or consequence in the pathogenesis of endometriosis [1,2,3]. According to some scientific studies, there is a link between an inflammatory process and oxidative stress which may contribute to the development of endothelial dysfunction [4]. Other research points to immunological dysfunction as an initiator of the disease [5]. Peritoneal inflammation observed in endometriosis may be connected with dysregulation of the hypothalamic-pituitary-adrenal axis as well. Inflammation itself can influence the expression of oestrogen receptors, which positively correlates with the expression of inflammatory cytokines in macrophages. Many of the mechanisms involved in the development of the disease are still awaiting elucidation. However, although inflammatory mediators are upregulated and inflammatory cells are activated, a pre-existing inflammation may not contribute to the development of endometriosis [6].

The Sampson’s retrograde menstruation theory introduced in 1927, the angiogenic and lymphogenic spread, and the metaplasia theory proposed in 1942, are not sufficient to provide a clear-cut explanation for all the manifestations of the disease. It is speculated that the polygenetic and polyepigenetic hypotheses, which have several clinical implications, are feasible enough to elucidate changes in the endometrium, immunology and placentation. A typical, deep and cystic ovarian endometriosis is often described as clonal in origin and manifested by clinical heterogeneity of the lesions, which may be suggestive of initial chromosomal modifications. Expression of the genetic changes transmitted at birth could increase predisposition towards endometriosis. New lesions may be formed throughout life due to cumulative genetic and epigenetic abnormalities. Bleeding, oxidative stress, body radiation and dioxins are regarded as additional factors for activation of this process. [7,8].

Endometriosis is characterized by the presence of endometrial-like tissue outside the uterus. The most common locations for ectopic endometrial implants are the ovaries, peritoneum and rectovaginal septum [9,10]. There are three types of endometriosis: peritoneal, ovarian and deeply infiltrating [11]. The incidence of endometriosis in women of reproductive age is between 6% and 10%. It is also estimated that endometriosis occurs in 21–47% of infertile women as well as in 71–87% of women experiencing chronic pelvic pain. Endometriosis is a major cause of markedly reduced quality of life in women suffering from this disease [10].

The most common symptoms of endometriosis are painful sexual intercourse (deep dyspareunia), pain before and/or during menstruation (dysmenorrhoea), pain when urinating (dysuria) and chronic pelvic pain [10,12]. Progression of the disease does not correlate with the aggravation of the symptoms and none of them are specific. Therefore, the time between the occurrence of the first symptoms and making the final diagnosis may be as long as 8–12 years [12]. As a result of the improper verification process, the patient may be unnecessarily treated for diseases that may mimic the symptoms associated with other chronic pain-related disorders, e.g., irritable bowel syndrome and pelvic inflammatory disease [13,14]. In addition, women with endometriosis experience a number of non-clinical symptoms that include depression, fatigue, and the feeling of isolation. Endometriosis has a negative impact on the psychological and social welfare of the patient [15].

Laparoscopy, an invasive method of examination, preceded by a transvaginal ultrasound and pelvic magnetic resonance imaging (MRI), is still regarded as the gold standard in the diagnosis of endometriosis [14]. Therefore, it is hoped that development of non-invasive diagnostic tools, such as ‘biomarkers’, could significantly reduce the time taken to diagnose endometriosis and enable monitoring the progression of the disease and the effectiveness of its treatment [13]. Replacing the invasive diagnostic methods with biomarkers which comply with the predetermined criteria, i.e., 94% sensitivity and 79% specificity, could be of considerable clinical usefulness [16,17].

This paper’s aim is to present and discuss the current status of biomarkers of endometriosis in the serum and urine. In our review, we focused on the main groups of markers which are: glycoproteins, growth factors, peptides, immunological markers, markers of oxidative stress, microRNAs (miRNAs) and long non-coding RNAs (lncRNAs) (Figure 1). This article also attempts to identify potential non-invasive biomarkers or sets of biomarkers which can be used in asymptomatic patients in the early stages of the disease.

We conducted a comprehensive literature review using electronic databases such as Pubmed, Science Direct and Google Scholar, and took into account articles published in English between 1988 and April 2021. Keywords such as: “biomarker”, “endometriosis”, “glycoproteins”, “urocortin”, “immunological markers”, “oxidative stress”, “microRNA”, “lncRNA”, “urinary biomarkers”, and various combinations of the above were used. Publications were selected if they related to the studies investigating potential biomarkers detected in women with endometriosis. In addition, we manually reviewed the references for each article to find potentially missed studies. As a result of this, we identified a total of 2073 relevant articles related to the topic of interest. Having considered the inclusion/exclusion criteria and eliminating duplicates, 93 studies were selected for analysis.

Inclusion criteria for the study selection:The samples of biomarkers could be collected from serum, plasma, whole blood, tissue, urineRandomized clinical trials, systemic reviews, meta-analysesAnimal, human, in vitro studies.

Exclusion criteria:Case reports, conference summaries, commentsInsufficient dataNot accessible as a full-text article for reviewLanguage other than EnglishStudies conducted on non-mammalian species.

The initial aim of this study was to identify potential novel diagnostic markers for the diagnosis of endometriosis, however, during the review process, it was suggested that the subject should be changed and urinary biomarkers and glycoproteins such as Cancer Antigen 125 (CA-125) were included.

## 2. Glycoproteins

Many studies have evaluated the usefulness of the serum glycoproteins as diagnostic tools in endometriosis. In medicine, they are commonly used for diagnosing and evaluation of malignancies [18].

CA-125 is a glycoprotein most commonly described as a potential marker for endometriosis. It is also known as Mucin-16 (the largest membrane-bound mucin) encoded by the MUC16 gene [19]. MUC16 is normally expressed by the epithelium of the ocular surface, upper respiratory tract, the mesothelium lining body cavities (pleural, peritoneal, and pelvic cavities), the internal organs and female reproductive tract, including the endometrium, fallopian tube and ovary [20]. It is worth noting that MUC16, as a membrane component of the non-receptive luminal uterine surface, prevents cell adhesion. The loss of MUC16 from the apical surface of the uterodome during the receptive phase of the reproductive cycle facilitates adhesion of the trophoblast and implantation of the embryo into the uterus [21].

MUC16 is a complex mucin consisting of three distinct domains: amino terminal, tandem repeat and carboxyl terminal. The extra-cellular fraction is secreted into the bloodstream and can be used as a biomarker to diagnose and monitor cancer. However, its role in tumorigenesis is not well understood. The large size of MUC16 and its extensive glycosylation, which results in its functional heterogeneity, render conducting research into this molecule difficult. Moreover, the lack of specific antibodies that detect the MUC16 domains, and the existence of splicing variants further impede insight into it [22,23]. Having said that, MUC16/CA-125 remains the best known tumour marker of the ovarian epithelial cells; however, it is nonspecific. Its elevated concentration is observed in patients with cancer of the breast, endometrium and lung, as well as in gastrointestinal and inflammatory conditions. An increased level of CA-125 is the most reliable marker for identification of epithelial ovarian cancer. Its suitability is also tested in endometriosis, an inflammatory disease in which CA-125 is secreted into the circulation by the endometrial and mesothelial cells [24,25,26].

To date, no clearly defined marker limit value has been determined. Most articles consider 35 U/mL as a cut-off point. It is assumed that the level of the marker is different in pre- and postmenopausal women, and the study of Karimi-Zarchi et al. showed that the best cut-off point was 37 U/mL in premenopausal women, whereas in postmenopausal women, it was 35 U/mL [27].

The studies investigating sensitivity for CA-125 provide different results [28,29,30]. However, although the CA-125 values fluctuate during different phases of the menstrual cycle, the value is usually higher during menstruation [25]. This is probably due to the increased inflammatory activity of the endometrial cells. It is suggested that the concentration of CA-125 should be tested in two phases of the cycle, i.e., in the middle of the cycle and in the menstrual phase. Positive results of CA-125 in the middle of the menstrual cycle will be indicative of a very high risk of endometriosis [25].

It has been scientifically proven that there is a correlation between high levels of CA-125 and the stage of endometriosis as well as its clinical type [31]. The sensitivity of endometriosis stages III and IV was 63.1%, compared to only 24.8% in stages I and II. Thus, investigation of the concentration of the marker may reveal a higher value in deeply infiltrating endometriosis with the presence of adhesions [32].

Currently, despite its relatively low sensitivity and specificity, CA-125 remains the only marker widely used in clinical practice in the diagnosis of endometriosis. To date, CA-125 has been used as a prognostic rather than a diagnostic marker. It is believed that the result of ≥35 U/mL in women with endometriosis symptoms may shorten the diagnosis time and bring with it an earlier implementation of the appropriate therapy [32,33].

However, simultaneous measuring of CA-125 concentration with other molecules yielded interesting results. Combinations of biomarkers showed different sensitivity and specificity for endometriosis. In order to diagnose endometriosis, Vodolazkaia et al. [34] used a biomarker model which included annexin V (a marker of apoptosis, considered as a molecule for diagnosis of minimal-mild endometriosis), vascular endothelial growth factors (VEGF), CA-125 and glycodelin/soluble intercellular-adhesion molecule-1 (sICAM-1). A multivariate analysis of these biomarkers enabled diagnosis with a sensitivity of 81–90% and a specificity of 63–81%, which gives a better diagnostic performance than the diagnostic performance of any single biomarker in the presented study. In 2019, Dorien et al. [35] conducted a study in which previously described diagnostic models for endometriosis could not be validated. Only CA-125 was retained in the models throughout the technical verification and validation study. Mihalyi et al. [36] found that the plasma levels of interleukin 6 (IL-6), IL-8 and CA-125 were elevated in all women with endometriosis and in those with minimal-mild endometriosis with a sensitivity of 87% and specificity of 71%. Irungu et al. [37] suggest that CA-125 has the greatest value when combined with follistatin and sICAM1, providing a sensitivity of 67% at 80% specificity.

CA 19-9 is a tumour marker which has been used especially in the diagnosis of gastrointestinal cancers. When it became clear that the endometrium also produces CA 19-9, researchers began to look for its application in diagnosing endometriosis. However, their results are strongly controversial [16,38]. One scientific study found that CA 19-9 was not related to endometriosis [38], while other researchers observed increased levels of this marker in women with advanced stages of endometriosis [25]. When compared to CA-125, its specificity and sensitivity were equal to 86–89% and 52–61%, respectively [39].

Other glycoproteins that were taken into consideration and reported in the specialist literature were: CA 15-3, CA 72-4, α-fetoprotein (AFP) and carcinoembryonic antigen (CEA). However, these results indicated little diagnostic value with regard to endometriosis.

Glycodelin A (GdA) is a glycoprotein known as an endometrial progesterone-related protein, produced mainly by the endometrial glands during the secretory phase of the cycle. It has a strong immunosuppressive function and is involved in the angiogenic and apoptotic processes. An increased concentration of GdA has been observed in the peripheral blood of women with endometriosis. This suggests that GdA plays a significant role in the process of endometriosis by promoting neovascularisation and cell proliferation in the formation of endometrial implants [40].

Kocbek et al. [41] evaluated GdA in the serum and peritoneal fluid concentrations in women with ovarian endometriosis. The sensitivity and specificity of GdA as a biomarker of ovarian endometriosis were 82.1% and 78.4% in the serum, and 79.7% and 77.5% in the peritoneal fluid. Moreover, it was noted that the severity and frequency of menstrual pain positively correlated with the concentration of GdA. Mosbach et al. [42] also observed significantly increased serum GdA levels in women with endometriosis in comparison to the control group. The sensitivity and specificity values were 91.7% and 75.0% for GdA in the serum, and 89.6% and 90.0% in the peritoneal fluid, respectively. The IL-6 levels were also found to be increased in this study. In the research, a sensitivity of 93.8% and specificity of 90% were recorded in the serum, and 85.4% and 89%, respectively, were recorded in the peritoneal fluid. Moreover, a significant correlation was observed between the IL-6 and GdA levels in the serum and peritoneal fluid and the disease progression.

Kalyani et al. [43] constructed an electrochemical immunosensor for ultra-sensitive detection of the novel antigen glycodelin. This method has potential for further development in clinical practice to diagnose stage I and/or II endometriosis. Its advantages are simple measurement of glycodelin levels in a clinical sample, low-cost construction and high sensitivity. Its accuracy is comparable with the enzyme-linked immunosorbent assay (ELISA).

The scientific studies presented above indicate that GdA has the potential to become a useful diagnostic marker for endometriosis.

## 3. Growth Factors and Peptides: Urocortin, Activin, Follistatin

To date, few studies have evaluated the use of urocortin as a diagnostic marker of endometriosis. Urocortin is a member of the corticotrophin-releasing hormone (CRH) family and is produced by the eutopic and ectopic endometria. It is also believed to play an important role in decidualization, which is an essential process during early pregnancy [44]. Another effect of the urocortin is mediation in the process of mast cell degranulation and an ability to increase the permeability of blood vessels [45]. There are three types of urocortin, i.e., Ucn1, Ucn2, Ucn3, which interact with two types of the CRH receptor. Ucn1 binds type 1 and type 2 CRH receptors, while Ucn2 and Ucn3 selectively bind to CRH-R2 [46].

According to some studies, evaluating the plasma urocortin levels enables detection of symptomatic endometriosis with a high sensitivity (76–88%) and specificity (88–90%) [47,48].

Maia et al., conducted a study in infertile women and/or those in chronic pelvic pain [47]. The purpose of the study was to assess the predictive value of Ucn1 in the detection of endometriosis in women with the above-mentioned symptoms. Women with symptomatic endometriosis had higher levels of Ucn1 (median 59 pg/mL, interquartile interval 48–107 pg/mL) compared to women with no lesions (median 34 pg/mL, interquartile interval 22–43 pg/mL). Moreover, women with disorders other than endometriosis also had elevated urocortin levels, but to a lesser extent. The foregoing results show that an increase in the plasma Ucn1 >46 pg/mL allows to differentiate between the occurrence of symptomatic endometriosis and endometriosis with no lesions (76% sensitivity, 88% specificity). However, it is not possible to distinguish endometriosis from other diseases (including ovarian teratoma, ectopic pregnancy, uterine leiomyoma). Nevertheless, it should be emphasized that the highest detection rate of endometriosis occurred in women who suffered from both infertility and chronic pelvic pain.

A link is also suspected between variations in the CRH/Ucn1 levels and progesterone resistance, which may be explained by the lack of growth of Ucn1 and CRH mRNA levels during the secretory phase of the menstrual cycle in women with endometriosis. Novembri et al., demonstrated that the expression of Ucn1 and CRH mRNA in healthy women was higher in the secretory phase compared to the proliferative phase, while in women with endometriosis, it was the same in both phases [44].

According to Florio et al., the urocortin levels in women with endometriosis were twice as high as in women with non-endometrial ovarian cysts (median 49 pg/mL, interquartile interval 41–63 pg/mL vs. median 19 pg/mL, interquartile interval 15–23 pg/mL) and they were significantly higher in the cystic content of endometriomas compared to the peritoneal fluid and plasma [48]. Elevated urocortin levels were indicative of endometriosis with 88% sensitivity and 90% specificity, while CA-125 was able to detect only 65% of the cases with the same specificity.

Nevertheless, not all of the reported scientific studies confirm the usefulness of urocortin as a marker of symptomatic endometriosis [49,50]. The studies comparing the level of urocortin in women with endometriosis and ovarian teratomas as well as between endometriosis and benign ovarian cysts revealed no significant differences.

Activin A is a growth factor belonging to the transforming growth factor β (TGF-β) family. Physiologically, it is produced by the healthy endometrium and its expression reaches peak values in the secretory phase of the menstrual cycle [51]. Activin A promotes the process of decidualization and is also believed to play a role in the immunological processes of the cells involved in the pathogenesis of endometriosis. It has been noticed that in endometriosis, the level of activin A increases both in the eutopic and ectopic endometria. The greatest increase was observed in ovarian endometrioma in comparison with the other types of endometriosis, but in comparison to the controls, its growth was not sufficient enough to be used as a marker [52].

Follistatin is an extracellular glycoprotein secreted at a constant level throughout the whole menstrual cycle and its growth is observed during early pregnancy. Its main action is neutralization of activin A, which leads to inhibition of the decidualization process [53]. The greatest increase in the plasma follistatin level was observed in the ovarian and peritoneal forms in patients with deep infiltrating endometriosis (DIE) as well as in healthy controls, which excludes its use as a marker of endometriosis [52].

Combinations of activin A and follistatin as markers of endometriosis showed the highest effectiveness. In this case, a significant increase in the ovarian form was observed, but it was not suitable to differentiate the other forms of endometriosis from healthy controls [52].

## 4. Immunological Markers

There are many indications that dysfunction of the immune system is involved in the pathogenesis of endometriosis. Many studies have been conducted to determine whether different populations of immune cells could be used as non-invasive markers of endometriosis.

Macrophages are one of the cells found in significant amounts in the peritoneal fluid. They are responsible for ectopic endometrial cell adhesion, implantation and growth. Moreover, macrophages secrete numerous substances which are said to influence the development of endometriosis [54].

Macrophages are considered to be the source of VEGFs in women with endometriosis. VEGF is responsible for angiogenesis in the endometrial tissue, which allows it to regenerate after menstruation, but also affects newly formed vessels. Studies in mice showed that after implantation of uterine tissue into the peritoneum, macrophages activation and increased VEGF secretion in response to the tumour necrosis factor α (TNF-α) and interleukin 6 (IL-6) occurred [55]. According to some studies, the level of TNF-α increased in patients with endometriosis and correlated with the severity of the disease [56].

Research into the macrophage migration inhibitory factor (MIF) has shown that it is a cytokine with strong immune-regulatory potential, affecting angiogenesis and tissue remodelling [57]. MIF has been observed to significantly increase its levels in endometrial lesions, especially in advanced stages of endometriosis [58].

Natural Killer (NK) cells may play an important role in the pathogenesis of endometriosis. They are believed to be responsible for the clearance of regurgitated endometrial cells from the peritoneal cavity. It has been observed that patients with endometriosis have reduced NK cell cytotoxicity. This suggests that NK cell dysfunction may allow implantation of endometrial cells into the peritoneal cavity and lead to endometriosis [59]. IL-12 may inhibit the process of endometriosis by the activation of NK cells [60]. It has also been demonstrated that the abnormal human leukocyte antigen (HLA) classes I and II expression leads to a decrease in their cytotoxic activity [61].

A compound such as sICAM-1 should also be distinguished as a marker with valuble potential. It is associated with reduced cytotoxic activity of NK cells. sICAM-1 is supposed to be relevant to implantation disorders and formation of endometrial lesions [54]. Matalliotakis et al. observed that the level of sICAM-1 was higher in women suffering from endometriosis infertility compared to healthy controls [62].

Kuessel et al., compared the serum vascular cell adhesion molecule-1 (sVCAM-1) levels between women with endometriosis and control subjects [63]. The serum concentration of sVCAM-1 was significantly higher in patients with endometriosis. Moreover, this result was not associated with lesion entity, disease severity, the phase of the menstrual cycle or cigarette smoking. In addition, they found that women with endometriosis had lower serum levels of sICAM-1 compared to the controls. However, the receiver operating characteristics (ROC) analysis investigating the feasibility of using sICAM-1 for diagnosing endometriosis revealed only a moderate predictive power, which leads to the assumption that the serum sICAM-1 is not a promising stand-alone biomarker for predicting endometriotic lesions. Nevertheless, measuring sICAM-1 could be of value for diagnosing endometriosis in combination with measuring sVCAM-1, thereby increasing its predictive power. Kuessel et al. concluded that the sVCAM-1/sICAM-1 ratio seems to be a better diagnostic tool for endometriosis than the individual marker [63].

Additionally, elevated monocyte chemotactic protein-1 (MCP-1) values were observed in the peritoneal fluid and plasma in women with endometriosis, especially in the early stages of the disease. Another study revealed its elevated values in the more advanced stages [16].

According to Cho et al., the use of neutrophil/lymphocyte ratio can be applied as a diagnostic method for endometriosis [64]. They observed that in women with endometriosis, neutrophilia might coexist with lymphocytopenia. The combined use of neutrophil/lymphocyte ratio and CA-125 concentration demonstrated a high sensitivity for endometriosis detection with a sensitivity of 69.3% and a specificity of 83.9% [64].

## 5. Oxidative Stress Markers

The formation of reactive oxygen species (ROS) is a physiological process regulated by antioxidant defense mechanisms. An imbalance between these two formations is called oxidative stress. Its significant role has been demonstrated in the inflammatory response of many diseases, including endometriosis [17]. It seems that as a result of retrograde menstruation, the endometrial cells and desquamated menstrual cells migrate into the peritoneal cavity, thereby inducing a chronic inflammatory response. This stimulates proinflammatory cytokines to activate immune cells such as granulocytes and macrophages, which are known to be capable of producing ROS [65].

Inadequate metabolism of free radicals and ROS has a significant impact on the use of thiols and carbonyls which seem to be associated with endometriosis and subfertility [66]. These markers were distinguished in a meta-analysis from the Cochrane database, with carbonyls, showing a sensitivity of 94% and specificity of 51% in the detection of endometriosis (cut-off point < 14.9 mM), as well as in thiols, and they were regarded as substances particularly useful in the detection of pelvic endometriosis, showing a sensitivity of 73% and a specificity of 80% (cut-off < 396.44 mM). Paraoxonase-1 (PON-1) was also found to be helpful in the diagnosis of pelvic endometriosis with a sensitivity of 98% and a specificity of 80% (cut-off < 141.54 U/mL) [67]. A review conducted by Carvalho et al. took into account 19 studies investigating oxidative stress markers in endometriosis. Eleven of them showed a significant increase in the marker levels in patients with endometriosis in comparison to healthy women [65]. There is also a study evaluating a correlation between the concentration of oxidative stress markers and severity of endometriosis. According to the analysis, the levels of glutathione peroxidase and superoxide dismutase significantly decreased in the severe stage of endometriosis, while the levels of lipid peroxide increased together with the severity of the disease [68]. However, other studies have shown that there is no correlation between endometriosis and the presence of oxidative stress markers [66].

Due to the multitude of factors regulating the level of oxidative stress, researchers emphasize the necessity for conducting further scientific studies to investigate if it is possible to use oxidative stress markers as diagnostic tests for endometriosis [17,69].

## 6. MiRNAs and lncRNAs

MiRNA is a small non-coding RNA molecule containing about 22–24 nucleotides. Its main function is the regulation of gene expression, it also affects processes of proliferation, differentiation, growth and apoptosis. MiRNAs are regulatory molecules that control expression of many genes and play key roles in many biological processes [70]. However, dysregulation of miRNA has been associated with many diseases, including endometriosis. MiRNA has brought a new perspective to the field of serum markers and become the subject of many research papers [71,72].

In 2020, Zhang et al., selected and tested specific types of miRNA: miR-134-5p, miR-197-5p, miR-22-3p, miR-320a, miR-494-3p, and miR-939-5p [72]. Two types of miRNA, i.e., miR-22-3p and miR-320a, were distinctly upregulated in the group of endometriosis patients in comparison to the control group. Moreover, a significant difference was noticed between the studied patients in stages I–II compared to stages III–IV. Thus, it seems that these two types of miRNA could be potential biomarkers for endometriosis [72].

In 2016, Cosar et al., evaluated expression of different miRNAs and concluded that out of all the tested samples, the miR-125b-5b level was significantly upregulated in women with endometriosis [73]. Importantly, miR-125b has been observed to affect the levels of TNF-α, IL-6, and IL-1β, which belong to the group of inflammatory cytokines, and their concentration is elevated in women with endometriosis [74]. Another factor that affects inflammatory cytokine levels is miR-20a. It may be potentially used as a marker in the early stage of endometriosis due to its effect on the levels of TGF-β and IL-8. In patients with endometriosis, miR-20a expression is downregulated, which leads to increased levels of aforementioned pro-inflammatory cytokines [40].

Moreover, Hudson et al., showed that miR-154-5p alone, or in combination with miR-196b, miR-378-3p, and miR-33a-5p, is linked to endometriosis [75].

Anastasiu et al. [40] as well as Bjorkman and Taylor [74] indicate the miR-200 family as a promising marker of endometriosis. The miR-200 family includes miR-200a, miR-200b and miR-141. Scientific studies have shown a reduced expression of miRNAs in a group of patients with endometriosis, and the studies cited by Anastasiu et al., are characterized by a sensitivity of 84.4% and s specificity of 66.7% [40,74].

In 2020, Moustafa et al., selected six miRNAs: miR-125b, miR-150, miR-342, 451a, miR-3613, Let-7b [75]. Regardless of the studied group’s diversity when it comes to the disease progression, menstrual cycle phase, different racial demographics and the presence of hormonal treatment, the obtained results showed a significant increase in the levels of miR-125b, miR-150, miR-342, 451a as well as a significant decrease in the levels of miR-3613 and Let-7b in patients with endometriosis. These results were characterized by a sensitivity of 83% and a specificity of 96%. In addition, the authors observed differences in miRNAs for different stages of the disease; they also noticed that the phase of the menstrual cycle and hormone treatment had no effect on the measured values. One of the most important conclusions of this research is that endometriosis can be distinguished from other gynecological pathologies on the basis of miRNAs [75].

lncRNAs, a member of the RNA family, is considered as a marker of endometriosis. lncRNAs are molecules longer than 200 nucleotides which, even though they do not code for proteins, can regulate gene expression directly or indirectly by affecting the expression of miRNAs [76]. Anastasiu et al., selected a panel of lncRNAs (NR_038395, NR_038452, ENST00000482343, ENST00000544649, and ENST00000393610) whose levels showed a significant dysregulation in the group of patients with endometriosis. Researchers also identified the lncRNA molecule TC0101441 whose increased level correlated with infertility, chronic pelvic pain, and recurrence of the disease [40].

MiRNAs as well as lncRNAs are very attractive diagnostic markers due to their lower complexity, tissue specificity, lack of known post-translational modifications and stability in blood, urine or tissues [77]. Researchers have summarized their studies on miRNA with a conclusion that it is an endometriosis marker with an average value of sensitivity of 86% and specificity of 88%, while in the case of lncRNA the sensitivity was 89.7% and specificity was 73.2% [78,79]. Although the results are promising, research into miRNA and lncRNA remains a new field of study and requires confirmation and further investigation [40,72,78].

## 7. Urinary Biomarkers in Endometriosis

According to the currently available scientific knowledge, biomarkers for endometriosis can also be detected in urine. Great advantages of using a urinary test in the diagnosis of endometriosis include its low cost, non-invasiveness and the fact that a urine sample can be collected by the patient herself. However, the reliability of laboratory techniques and changing levels of the urinary biomarker during the menstrual cycle are limitations of the test.

Enolase I (NNE) is an enzyme detected in urine and has been studied as a biomarker in patients with endometriosis. The urinary NNE expression corrected for the creatinine ratio (NNE-Cr) was significantly higher in patients with endometriosis. Having analysed the results, the researchers concluded that NNE cannot be a stand-alone diagnostic marker for endometriosis. They suggest that it may have diagnostic value when combined with the serum CA-125. However, this area of interest requires further research investigating patients with a broader spectrum of endometriosis [80,81].

Cho et al. [82] evaluated the diagnostic efficacy of the urinary vitamin D-binding protein (VDBP). The urinary VDBP levels corrected for creatinine (VDBP-Cr) expression were elevated in patients with endometriosis. The sensitivity of this biomarker was 58% and the specificity was 76%. VDBP-Cr was only different in the luteal phase of the cycle in women with endometriosis compared to the control group, which may be a limitation of the study. More in-depth evaluation of VDBP across the spectrum of endometriosis, especially in the luteal phase, is needed. An important aspect of further research is to determine the role, if any, that VDBP plays in endometriosis [81].

With the use of proteomic techniques such as MALDI-TOF MS (matrix-assisted laser desorption/ionization mass spectrometry (MALDI MS) and time-of-flight analyzer (TOF)), different peptide markers were described in the urine of women with endometriosis and compared with the controls. El-Kasti et al. [83] evaluated the diagnostic accuracy of two peptides identified by their mass profile. The preovulatory peptide mass of 1767.1 showed a sensitivity of 75% and a specificity of 85%. The luteal peptide mass of 1824.3 Da showed a sensitivity of 77% and a specificity of 73%. Wang et al. [84] developed a genetic algorithm that included five peptides with masses of 1433.9 Da, 1599.4 Da, 2085.6 Da, 6798.0 Da, and 3217.2 Da. This model showed a sensitivity of 91% and a specificity of 93%. These results are promising but require further validation in large populations presenting all stages of endometriosis.

Another protein obtained from urine and considered as a biomarker for endometriosis is cytokeratin 19 (CK 19). Previous studies [85,86] have shown no differences between women with endometriosis and healthy population. However, Tokushige et al. [87] used proteomic techniques and thus demonstrated the presence of CK-19 in urine in women with endometriosis, comparing with a control group. In this survey, the study group is not big enough to conclusively specify the value of CK-19 as biomarker for endometriosis. It is necessary to perform more large-scale studies involving more patients. Presumably, CK 19 is not reliable as a diagnostic biomarker, as confirmed by Liu et al. [81].

Proestling et al. [88] investigated the usefulness of cell adhesion molecules such as sVCAM-1, sICAM-1, E-selectin and P-selectin as urinary biomarkers in the diagnosis of endometriosis. However, they found no significant differences in their urinary levels between women with and without endometriosis.

Chen and et al. [89] conducted a study to identify novel protein biomarkers that can be used to diagnose endometriosis. For the first time, histone 4 was identified as a potential biomarker for endometriosis with a sensitivity of 70% and specificity of 80%. This study may provide a new direction in the search for the most appropriate biomarker for endometriosis.

## 8. Perspectives

Endometriosis is a disease with both physical and psychological consequences. It can affect fertility and chances of having a baby as well as getting an education or a steady job, it can also reduce the quality of a woman’s social life and her physical activity. Over recent decades, endometriosis has been associated with the risk of several chronic diseases such as cancer, cardiovascular and autoimmune diseases. As the mechanisms of endometriosis formation are still unclear [1,2,3,4,5], there is a need to continue the search for novel diagnostic methods that can detect the disease at its early stage.

Due to the lack of effective, non-invasive methods, endometriosis diagnostics is a long process. Therefore, the development of new and more effective methods could speed it up and enable the detection of this disease in its early stages. The reviewed literature suggests that biomarkers which can be detected in the blood serum and urine seem to be promising. Markers detected in the urine, such as NNE-Cr, VDBP-Cr and histone 4, are particularly interesting because of their non-invasive nature. The suggested diagnostic solutions might be readily accepted by patients since they are painless and can be performed in the privacy of their own home. In addition, they could be easy to handle and interpret. The tests would involve collecting a urine sample into a cup and pouring a small amount of the fluid into a special container. After the recommended time, the test result would appear as a colour change or designated symbol. This non-invasive rapid form of initial diagnosis of endometriosis could be used especially among asymptomatic patients with a family history of history of the disease. It is important to note that these biomarkers are still in the research phase.

In our opinion, another alternative diagnostic option may be a combination of glycoproteins, growth and immunological factors such as CA-125 + follistatin + sICAM1 or activin A + follistatin derived from the patient’s blood serum (Figure 2). In a review paper of Niseblat et al., a combination of VDBP-Cr (urine) and Ca-125 (serum) for the diagnosis of pelvic endometriosis was also highlighted [90]. It presented a 74% sensitivity and 97% specificity and met the criteria of a potential triage test (cut-off point >2755). This combination proved to be more effective in the diagnosis of endometriosis compared to VDBP only (sensitivity 58%, specificity 55%). However, further studies are needed to evaluate its usefulness in the diagnostic process of the disease [90]. While analyzing the available data, it is worth mentioning the genetic background of the disease. Several clinical trials investigated the use of miR-134-5p, miR-197-5p, miR-22-3p, miR-320a, miR-494-3p, miR-939-5p as potential biomarkers in endometriosis. MiRNAs and lncRNAs, due to the tissue specificity, lack of known post-translational modifications and stability in the blood and urine, may be promising diagnostic factors for women affected with endometriosis.

Nevertheless, testing a genetic and immunological material is time-consuming, requires experienced personnel and financial resources. It is worth mentioning that due to the specific nature of the tissue itself, the gene expression is constantly changing. Therefore, it may happen that in a given sample there is a high expression of the studied gene and the product of that expression has not appeared yet. Over time, as the method becomes better known and more popular, and the techniques for conducting the tests have been improved and new ones are developed, the genetic methods may become the future of endometriosis diagnostics and an opportunity for many patients.

Another important aspect that needs to be addressed in this paper is a rigorous assessment of diagnostic tests’ credibility in our clinical practice. In connection with this, evaluation of a diagnostic test requires some familiarity with the used measurements. Some authors suggested a predetermined sensitivity of 94% and specificity of 79% for a clinically useful blood test to replace diagnostic surgery for endometriosis [67,91]. Sensitivity and specificity are measures of diagnostic accuracy that are not affected by the prevalence of the condition.

Another important diagnostic set for endometriosis are the predictive values of a negative or positive test result [92]. The positive predictive value (PPV) is the proportion of those positively tested individuals who have the disease, while the negative predictive value (NPV) denotes the proportion of individuals tested negatively who do not have the disease.

Two other worthwhile measurements to assess the value of performing a diagnostic test are: likelihood ratios (LRs) and receiver operator characteristic curves (ROC) [92]. The LR is a ratio of the true positives (sensitivity) to the false positives (1-specificity) at a particular value of a test [93]. In these calculations, both sensitivity and specificity are integrated. It should also be pointed out that LR seems to be a more intelligible way of conveying the accuracy of a diagnostic test to the user. Bearing in mind that the application of LR enables a more appropriate interpretation of tests, it merits further adoption into clinical practice [94]. Alternatively, these quantities can be plotted for all observed values of a predictor or test and the generated curve (ROC). The area under the ROC curve (AUC) is a common measure of the diagnostic accuracy of a test. Calculating LR without the ROC curve can be misleading since similar LR values are possible with quite different ROC areas [94].

In a meta-analysis performed to assess the diagnostic performance of the serum CA125 in detecting endometriosis, the ROC curve showed a poor diagnostic performance [26]. At a specificity of 90%, a sensitivity of 28% was reported. If the sensitivity was increased to 50%, the specificity dropped to 72%.

The LRs for the peritoneal TNF-α in patients with endometriosis were calculated by Bedaiwy et al., The authors reported TNF-α concentration of 20 pg/mL, a 96% sensitivity and a 95% specificity (positive LR of 19.2 and negative LR of 0.04) at a cut-off peritoneal fluid [95]. Nevertheless, the authors concluded that this test may not be practical because it would require an intervention to obtain the peritoneal fluid.

## 9. Conclusions

Due to endometriosis heterogeneity, perfect diagnostic markers should give an array of results fulfilling the requirements of specificity and sensitivity at the same time. The presented paper shows that even though we have a number of promising markers, none of them meets the mentioned criteria on their own (Table 1).

It goes without saying that further studies are required to combine all of the available methods, including the serum and urinary markers, glycoproteins, growth factors and peptides, immunological markers, and also genomic technologies and non-invasive imaging methods such as ultrasonography or MRI. A variety of combined tests should help to unify the results and lead to making an early diagnosis, which would definitely improve the patients’ quality of life.

## Figures and Tables

**Figure 1 jcm-10-02762-f001:**
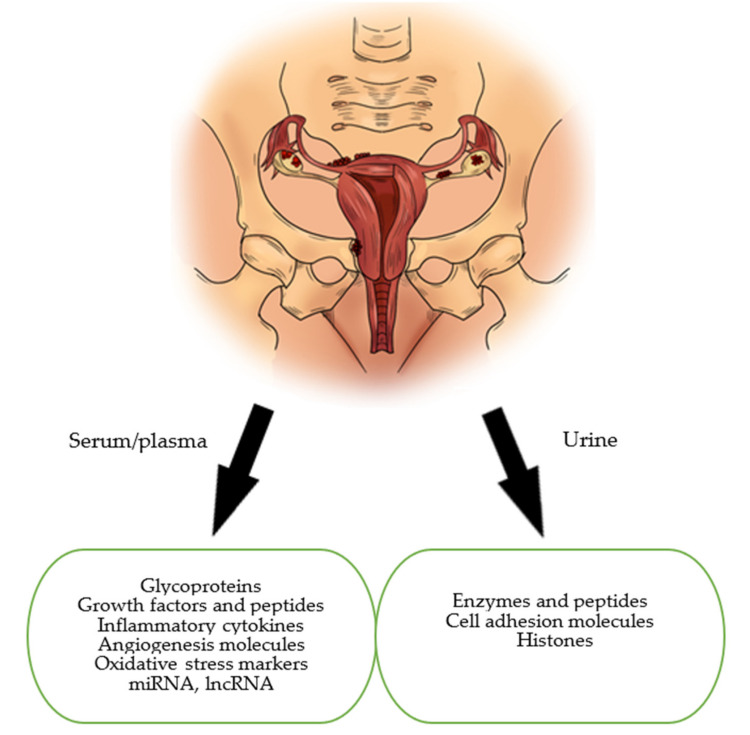
The most typical locations of ectopic endometriosis implants. Several potential endometriosis biomarkers are produced by the endometriosis implants themselves, by affected tissues and/or by the immune system.

**Figure 2 jcm-10-02762-f002:**
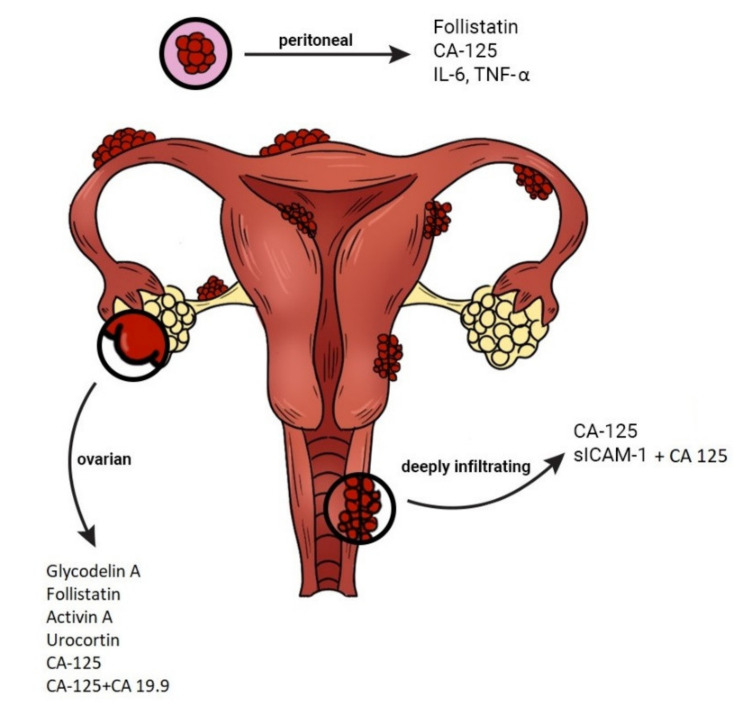
Potential biomarkers for the diagnosis of peritoneal, ovarian and deep infiltrating pelvic endometriosis. CA—Cancer Antigen; IL—interleukin; slCAM—soluble intercellular-adhesion molecule; TNF—tumour necrosis factor.

**Table 1 jcm-10-02762-t001:** Complete list of analysed biomarkers.

Biomarkers	Molecules	References
Glycoproteins	CA-125, CA 19.9, CA 15.3, CA 72, AFP, CEA, Glycodelin A	[18,19,20,21,22,23,24,25,26,27,28,29,30,31,32,33,34,35,36,37,38,39,40,41,42,43,44]
Growth factors	Urocortin, Activin A, Follistatin	[44,45,46,47,48,49,50,51,52,53]
Immunological markers	VEGF, TNF-α, IL-6, NK, slCAM-1, sVCAM-1, MCP-1	[16,54,55,56,57,58,59,60,61,62,63,64]
Oxidative stress markers	ROS	[17,65,66,67,68,69]
miRNA, lncRNA	miR-134-5p, miR-197-5p, miR-22-3p, miR-320a, miR-494-3p, miR-939-5p, NR_038395, NR_038452, ENST00000482343, ENST00000544649, and ENST00000393610, TC0101441	[40,70,71,72,73,74,75,76,77,78,79]
Urinary biomarkers	NNE-Cr, VDBP-Cr, CK-19, sVCAM-1, sICAM-1, E-selectin, P-selectin, Histone 4	[80,81,82,83,84,85,86,87,88,89]

AFP—α-fetoprotein; CA—Cancer Antigen; CEA—carcinoembryonic antigen; CK—cytokeratin; IL—interleukin; lncRNA—long non-coding RNAs; MCP—monocyte chemotactic protein; miRNA—microRNA; NK—Natural Killer; NNE-Cr—Enolase I expression corrected for the creatinine ratio; ROS—reactive oxygen species; TNF—tumour necrosis factor; slCAM—soluble intercellular-adhesion molecule; sVCAM—serum vascular cell adhesion molecule; VDBP-Cr—vitamin D-binding protein corrected for creatinine; VEGF—vascular endothelial growth factors.

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
