# Peer review of "The Potential of Non-Invasive Biomarkers for Early Diagnosis of Asymptomatic Patients with Endometriosis"

_jcm, 2021, doi:10.3390/jcm10132762_

Round 1
Reviewer 1 Report
The authors have addressed comments from the initial review. This manuscript is appropriate for publication.
Author Response
Dear Reviewer 1,
Thank you very much for finding the time to read our manuscript and make helpful remarks. We would like to take this opportunity to express our gratitude for all the previous valuable and useful comments which have substantially contributed to improving the quality of our paper.
Reviewer 2 Report
The manuscript of Kimber-Trojnar has improved, but several major topics need to addressed.
- I suggested to include a PRISMA flowchart for the literature search and data selection. However, the authors completely skipped their description of the search and selection strategy. Thus, the reader cannot judge the reliability of the review.
- Page 4, line 98. CA-125 which is also known as Mucin-16 does show a restricted protein abundance in ocular, nasal, tracheal and reproductive tract epithelia (normal tissues). I hope that I have not missed some. However, MUC16 is upregulated in many cancers which might be a problem for the use of CA-125 as a reliable marker for endometriosis besides other problems. CA-125 is a membrane-bound mucin from which the extracellular domain is shed and can be detected in blood. Severe problems for accurate detection are alternative splicing and heavy glycosylation. Furthermore, loss of MUC16 from the uterus facilitates binding of the blastocyst to the uterine epithelium. In my opinion these are important information the reader should get about CA125/MUC16.
- Page 4, line 145 Please replace divided by controversial
- Page 5 168 170-174 and 188. Please present the results from these studies with respect to the usefulness as a diagnostic test (e.g. PPV; NPV, Sensitivity, Specificity etc.)
- Page 8, line 324-370. Although I advised the authors to search for the work of Taylor, again they missed one very important one about miRNAs which was published in AJOG 2020;223(4):557.e1-557.e11. This is the first report that miRNAs can differentiate between endometriosis and other gynecological pathologies.
- Although some authors (e.g. Nisenblat et al., 2016 Cochrane Database Syst Rev 2016(5):CD012179) suggested a predetermined sensitivity of 94% and sensitivity of 79% for a clinically useful blood test to replace diagnostic surgery for endometriosis, however, the positive and negative likelihood ratios are also very good parameters to evaluate the value of performing a diagnostic test. In these calculations both sensitivity and specificity are integrated. However, the authors have never shown any likelihood ratios or have calculated them on their own from the published data. This would be helpful in getting an idea about the usefulness of the markers presented.
Author Response
Dear Reviewer 2,
We would like to resubmit our manuscript entitled “The potential of non-invasive biomarkers for early diagnosis of asymptomatic patients with endometriosis”. We appreciate your valuable remarks and hope that the quality of our manuscript is going to meet your expectations now that we have made some suggested alternations.
We have rearranged our paper in accordance with your valuable comments and suggestions. The manuscript has been checked by a native-speaker.
The manuscript of Kimber-Trojnar has improved, but several major topics need to be addressed.
Thank you for your valuable comments. We do hope that owing to your spot-on remarks the quality of our paper has improved.
1. I suggested to include a PRISMA flowchart for the literature search and data selection. However, the authors completely skipped their description of the search and selection strategy. Thus, the reader cannot judge the reliability of the review.
We agree wholeheartedly with the Reviewer`s opinion regarding inclusion of a PRISMA flowchart for the literature search and data selection. However, at this stage, after we have changed the title and aim of the manuscript, thereby extending the presented subject matter, which was indicated by one of the reviewers of the first revision, creation of a flowchart could be distorted and / or outdated. Nevertheless, we would like to thank you for drawing our attention to the usefulness of preparing a PRISMA flowchart - we will certainly bear it in mind while preparing future review articles.
- Page 4, line 98. CA-125 which is also known as Mucin-16 does show a restricted protein abundance in ocular, nasal, tracheal and reproductive tract epithelia (normal tissues). I hope that I have not missed some. However, MUC16 is upregulated in many cancers which might be a problem for the use of CA-125 as a reliable marker for endometriosis besides other problems. CA-125 is a membrane-bound mucin from which the extracellular domain is shed and can be detected in blood. Severe problems for accurate detection are alternative splicing and heavy glycosylation. Furthermore, loss of MUC16 from the uterus facilitates binding of the blastocyst to the uterine epithelium. In my opinion these are important information the reader should get about CA125/MUC16.
Thank you very much for your valuable and highly perceptive remarks. We have expanded the issue of CA125/MUC16 as follows:
“Cancer Antigen 125 (CA-125) is a glycoprotein most commonly described as a potential marker for endometriosis. It is also known as Mucin-16 (the largest membrane-bound mucin) encoded by the MUC16 gene [16]. MUC16 is normally expressed by the epithelium of the ocular surface, upper respiratory tract, the mesothelium lining body cavities (pleural, peritoneal, and pelvic cavities), the internal organs and female reproductive tract, including the endometrium, fallopian tube and ovary [17]. It is worth noting that MUC16, as a membrane component of the non-receptive luminal uterine surface, prevents cell adhesion. The loss of MUC16 from the apical surface of the uterodome during the receptive phase of the reproductive cycle facilitates adhesion of the trophoblast and implantation of the embryo into the uterus [18].
MUC16 is a complex mucin consisting of three distinct domains: amino terminal, tandem repeat and carboxyl terminal. The extra-cellular fraction is secreted into the bloodstream and can be used as a biomarker to diagnose and monitor cancer. However, its role in tumorigenesis is not well understood. The large size of MUC16 and its extensive glycosylation, which results in its functional heterogeneity, render conducting research into this molecule difficult. Moreover, the lack of specific antibodies that detect the MUC16 domains, and the existence of splicing variants further impede the insight into it [19,20]. Having said that, MUC16/CA-125 still remains the best known tumour marker of the ovarian epithelial cells, however, it is nonspecific.”
References:
16. Das, S.; Batra, S.K. Understanding the Unique Attributes of MUC16 (CA125): Potential Implications in Targeted Therapy. Cancer Res. 2015, 75, 4669-4674.
17. Bafna, S.; Kaur, S.; Batra, S.K. Membrane-bound mucins: the mechanistic basis for alterations in the growth and survival of cancer cells. Oncogene 2010, 29, 2893-2904.
18. Gipson, I.K.; Blalock, T.; Tisdale, A.; Spurr-Michaud, S.; Allcorn, S.; Stavreus-Evers, A.; Gemzell, K. MUC16 is lost from the uterodome (pinopode) surface of the receptive human endometrium: in vitro evidence that MUC16 is a barrier to trophoblast adherence. Biol. Reprod. 2008, 78, 134-142.
19. Haridas, D.; Ponnusamy, M.P.; Chugh, S.; Lakshmanan, I.; Seshacharyulu, P.; Batra, S.K. MUC16: molecular analysis and its functional implications in benign and malignant conditions. FASEB J. 2014, 28, 4183-4199.
20. Felder, M.; Kapur, A.; Gonzalez-Bosquet, J.; Horibata, S.; Heintz, J.; Albrecht, R.; Fass, L.; Kaur, J.; Hu, K.; Shojaei, H.; Whelan, R.J.; Patankar, M.S. MUC16 (CA125): tumor biomarker to cancer therapy, a work in progress. Mol. Cancer 2014, 13, 129.
- Page 4, line 145 Please replace divided by controversial
We have replaced “divided” by “controversial”
- Page 5 168 170-174 and 188. Please present the results from these studies with respect to the usefulness as a diagnostic test (e.g. PPV; NPV, Sensitivity, Specificity etc.)
Thank you very much for your perceptive comments. We have expanded these parts as follows:
The previous line 168:
“In the research, a sensitivity of 93.8% and specificity of 90% were recorded in the serum, and 85.4% and 89% , respectively, were recorded in the peritoneal fluid.”
The previous lines 170-174:
“Its accuracy is comparable with the enzyme-linked immunosorbent assay (ELISA).”
The previous line 168:
“According to some studies, evaluating the plasma urocortin levels enables detection of symptomatic endometriosis with high sensitivity (76-88%) and specificity (88-90%) [44,45].”
- Page 8, line 324-370. Although I advised the authors to search for the work of Taylor, again they missed one very important one about miRNAs which was published in AJOG 2020;223(4):557.e1-557.e11. This is the first report that miRNAs can differentiate between endometriosis and other gynecological pathologies.
We agree that this publication in AJOG 2020;223(4):557.e1-557.e11 is very important and valuable. In the previous revision, we added two papers co-authored by Prof. Taylor. In the current version we present the suggested report on miRNAs as follows:
“In 2020, Moustafa et al. selected 6 miRNAs: miR-125b, miR-150, miR-342, 451a, miR-3613, Let-7b [72]. Regardless of the studied group`s diversity when it comes to the disease progression, menstrual cycle phase, different racial demographics and the presence of hormonal treatment, the obtained results showed a significant increase in the levels of miR-125b, miR-150, miR-342, 451a as well as a significant decrease in the levels of miR-3613 and Let-7b in patients with endometriosis. These results were characterized by a sensitivity of 83% and specificity of 96%. In addition, the authors observed differences in miRNAs for different stages of the disease, they also noticed that the phase of the menstrual cycle and hormone treatment had no effect on the measured values. One of the most important conclusions of this research is that endometriosis can be distinguished from other gynecological pathologies on the basis of miRNAs [72].”
72. Moustafa, S.; Burn, M.; Mamillapalli, R.; Nematian, S.; Flores, V.; Taylor, H.S. Accurate diagnosis of endometriosis using serummicroRNAs. Am. J. Obstet. Gynecol. 2020, 557, 1-11
- Although some authors (e.g. Nisenblat et al., 2016 Cochrane Database Syst Rev 2016(5):CD012179) suggested a predetermined sensitivity of 94% and sensitivity of 79% for a clinically useful blood test to replace diagnostic surgery for endometriosis, however, the positive and negative likelihood ratios are also very good parameters to evaluate the value of performing a diagnostic test. In these calculations both sensitivity and specificity are integrated. However, the authors have never shown any likelihood ratios or have calculated them on their own from the published data. This would be helpful in getting an idea about the usefulness of the markers presented.
Thank you very much for your comments. Following your advice, we have presented these aspects in the section entitled “Perspectives”:
“Another important aspect that needs to be addressed in this paper is a rigorous assessment of diagnostic tests’ credibility in our clinical practice. In connection with this, evaluation of a diagnostic test requires some familiarity with the used measurements. Some authors suggested a predetermined sensitivity of 94% and sensitivity of 79% for a clinically useful blood test to replace diagnostic surgery for endometriosis [64,86]. Sensitivity and specificity are measures of diagnostic accuracy that are not affected by the prevalence of the condition.
Another important diagnostic set for endometriosis are the predictive values of a negative or positive test result [87]. The positive predictive value (PPV) is the proportion of those with a positive result that have the disease, whereas the negative predictive value (NPV) is the proportion of those with a negative test that do not have the disease.
Two other worthwhile measurements to assess the value of performing a diagnostic test are: likelihood ratios (LRs) and receiver operator characteristic curves (ROC) [87]. The LR is a ratio of the true positives (sensitivity) to the false positives (1-specificity) at a particular value of a test [88]. In these calculations both sensitivity and specificity are integrated. Alternatively, these quantities can be plotted for all observed values of a predictor or test and the generated curve (ROC). The area under the ROC curve (AUC) is a common measure of the diagnostic accuracy of a test. Calculating LR without the ROC curve can be misleading since similar LR values are possible with quite different ROC areas [89].
In a meta-analysis performed to assess the diagnostic performance of the serum CA125 in detecting endometriosis, the ROC curve showed a poor diagnostic performance [90]. At a specificity of 90%, a sensitivity of 28% was reported. If the sensitivity was increased to 50% the specificity dropped to 72%.
The LRs for the peritoneal TNF-alfa in patients with endometriosis were calculated by Bedaiwy et al. The authors reported TNF-alfa concentration of 20 pg/mL, a 96% sensitivity and a 95% specificity (positive LR of 19.2 and negative LR of 0.04) at a cut-off peritoneal fluid [91]. Nevertheless, the authors concluded that this test may not be practical because it would require an intervention to obtain the peritoneal fluid.”
References:
64. Nisenblat, V.; Bossuyt, P.M.; Shaikh, R.; Farquhar, C.; Jordan, V.; Scheffers, C.S.; Mol, B.W.; Johnson, N.; Hull, M.L. Blood biomarkers for the non-invasive diagnosis of endometriosis. Cochrane Database Syst. Rev. 2016, 5, CD012179.
86. Gupta, D.; Hull, M.L.; Fraser, I.; Miller, L.; Bossuyt, P.M.; Johnson, N.; Nisenblat, V. Endometrial biomarkers for the non-invasive diagnosis of endometriosis. Cochrane Database Syst. Rev. 2016, 4, CD012165. 87. Falcone, T.; Mascha, E. The elusive diagnostic test for endometriosis. Fertil. Steril. 2003, 80, 886-888. 88. Hartzes, A.M.; Morgan, C.J. Meta-analysis for diagnostic tests. J. Nucl. Cardiol. 2019, 26, 68-71. 89. Florkowski, C.M. Sensitivity, specificity, receiver-operating characteristic (ROC) curves and likelihood ratios: communicating the performance of diagnostic tests. Clin. Biochem. Rev. 2008, 29 Suppl 1, S83-S87.
90. Mol, B.W.; Bayram, N.; Lijmer, J.G.; Wiegerinck, M.A.; Bongers, M.Y.; van der Veen, F.; Bossuyt, P.M. The performance of CA-125 measurement in the detection of endometriosis: a meta-analysis. Fertil. Steril. 1998, 70, 1101-1108.
91. Bedaiwy, M.A.; Falcone, T.; Sharma, R.K.; Goldberg, J.M.; Attaran, M.; Nelson, D.R.; Agarwal, A. Prediction of endometriosis with serum and peritoneal fluid markers: a prospective controlled trial. Hum. Reprod. 2002, 17, 426-431.
We would like to take this opportunity to thank you for all the valuable and highly perceptive remarks which have definitely made a substantial contribution to the quality of our paper.
Yours faithfully,
Assoc. Prof. Zaneta Kimber-Trojnar, M.D., Ph.D.
Chair and Department of Obstetrics and Perinatology
Medical University of Lublin
Jaczewskiego 8, 20-090 Lublin, Poland
Tel: +48 81 7244 769;
Fax: +48 81 7244 841
E-mail: zkimber@poczta.onet.pl

Reviewer 3 Report
The authors have adapted their manuscript according to all the comments in an intelligent way. As a result, the paper is very easy to read for a general audience. The authors have also added very recent studies with specific technical information, further increasing the value of the present review for a specialized readership. The new figures and tables are helpful and "sexy".
I have very narrow comments (on an editorial point of view) to keep improving this manuscript:
- Some sentences are overly cautious (ie. "may suggest" could be simplified in "suggests", line 157).
- Please, correct reference 27 "FO Dorien" (also line 134).
Author Response
Dear Reviewer 3,
"The authors have adapted their manuscript according to all the comments in an intelligent way. As a result, the paper is very easy to read for a general audience. The authors have also added very recent studies with specific technical information, further increasing the value of the present review for a specialized readership. The new figures and tables are helpful and "sexy"."
Thank you very much for your comments. We would also like to express our gratitude to the Reviewer for all the previous valuable and helpful comments which have made a substantial contribution to the quality of our paper.
"I have very narrow comments (on an editorial point of view) to keep improving this manuscript:
- Some sentences are overly cautious (ie. "may suggest" could be simplified in "suggests", line 157)."
Thank you very much for your comments. Among others, we have replaced “may suggest” with “suggests”.
- "Please, correct reference 27 "FO Dorien" (also line 134)."
We have corrected the author’s name in the line 134 and reference 27.
Round 2
Reviewer 2 Report
There are still several issues whivh should be changed:
Page 1, lines 35-39 Nevertheless, Sampson's theory of transplantation and implantation...appear to point to the presence of a chronic inflammatory response, both local and systemic, as a factor worth emphasizing.
As already stated in my first review, this was never stated by Sampson or some of the other people. For the involvement of inflammation in the pathogenesis other manuscripts have to be cited.
Page 1 line 39-40 All the hypotheses mentioned above take into account the involvement of numerous interactions at the hormonal, genetic, immunological, and environmental levels.
Again this is not true for Sampson, not for Waldeyer's and Meyer’s coelomic metaplasia theory, and also not for Halban’s lymphovascular microembolization theory. To the best of my knowledge the genetic/epigenetic hypothesis by Koninckx et al. 2019 in Fertil Steril (and some earlier references by him and his group) is best describing the hormonal, genetic, immunological, and environmental interactions.
Still, no search strategy is included and no inclusion/exclusion criteria are mentioned. Although you changed your topic considerably after my first review, the reader is interested also in the changed search strategy.
In the paragraph about urine biomarkers you missed Tokushige et al. 2011 in Fertil Steril with keratin 19 and also the reference of Liu et al. 2015 in Cochrane could be included.
In the manuscript by Florkowski 2008 (ref. no. 89), the author also mentioned that the likelihood ratio is more intelligible to users. Thus, I would like to know whether it is possible to include it also in the present review.
Author Response
Dear Reviewer 2,
We appreciate your valuable remarks and hope that the quality of our manuscript is going to meet your expectations now that we have made some suggested alternations.
There are still several issues which should be changed:
Page 1, lines 35-39 Nevertheless, Sampson's theory of transplantation and implantation...appear to point to the presence of a chronic inflammatory response, both local and systemic, as a factor worth emphasizing.
As already stated in my first review, this was never stated by Sampson or some of the other people. For the involvement of inflammation in the pathogenesis other manuscripts have to be cited.
We apologize for the confusion caused. Thank you very much for your highly perceptive remarks. We have expanded the issue of the involvement of inflammation as follows:
“According to some scientific studies, there is a linkage between an inflammatory process and oxidative stress which may contribute to the development of endothelial dysfunction [4]. Other research points to immunological dysfunction as an initiator of the disease [5]. Peritoneal inflammation observed in endometriosis may be connected with dysregulation of the hypothalamic-pituitary-adrenal axis as well. Inflammation itself can influence the expression of oestrogen receptors, which positively correlates with the expression of inflammatory cytokines in macrophages. Many of the mechanisms involved in the development of the disease are still awaiting elucidation. However, although inflammatory mediators are upregulated and inflammatory cells are activated, a pre-existing inflammation may not contribute to the development of endometriosis [6].”
References:
“4. Samimi, M.; Pourhanifeh, M.H.; Mehdizadehkashi, A.; Eftekhar, T.; Asemi, Z. The role of inflammation, oxidative stress, angiogenesis, and apoptosis in the pathophysiology of endometriosis: Basic science and new insights based on gene expression. J. Cell Physiol. 2019, 234, 19384-19392.
- Symons, L.K.; Miller, J.E.; Kay, V.R.; Marks, R.M.; Liblik, K.; Koti, M.; Tayade, C. The Immunopathophysiology of Endometriosis. Trends Mol. Med. 2018, 24, 748-762.
- Jiang, L.; Yan, Y.; Liu, Z.; Wang, Y. Inflammation and endometriosis. Front. Biosci. (Landmark Ed), 2016, 21, 941-948.”
Page 1 line 39-40 All the hypotheses mentioned above take into account the involvement of numerous interactions at the hormonal, genetic, immunological, and environmental levels.
Again this is not true for Sampson, not for Waldeyer's and Meyer’s coelomic metaplasia theory, and also not for Halban’s lymphovascular microembolization theory. To the best of my knowledge the genetic/epigenetic hypothesis by Koninckx et al. 2019 in Fertil Steril (and some earlier references by him and his group) is best describing the hormonal, genetic, immunological, and environmental interactions.
Thank you for your valuable comments. We do hope that owing to your spot-on remarks the quality of our paper has improved.
“The Sampson's retrograde menstruation theory introduced in 1927, the angiogenic and lymphogenic spread, or the metaplasia theory proposed in 1942, are not sufficient to provide clear-cut explanation for all the manifestations of the disease. It is speculated that the polygenetic and polyepigenetic hypotheses, which have several clinical implications, are feasible enough to elucidate changes in the endometrium, immunology and placentation. A typical, deep and cystic ovarian endometriosis is often described as clonal in origin and manifested by clinical heterogeneity of the lesions, which may be suggestive of initial chromosomal modifications. An expression of the genetic changes transmitted at birth could increase the predisposition towards endometriosis. New lesions may be formed throughout life due to cumulative genetic and epigenetic abnormalities. Bleeding, oxidative stress, body radiation and dioxins are regarded as additional factors for activation of this process. [7,8].”
References:
- Koninckx, P.R.; Ussia, A.; Adamyan, L.; Wattiez, A.; Gomel, V.; Martin, D.C. Pathogenesis of endometriosis: the genetic/epigenetic theory. Fertil Steril. 2019, 111, 327-340.
- Gordts, S.; Koninckx, P.; Brosens, I. Pathogenesis of deep endometriosis. Fertil Steril. 2017, 108, 872.e1-885.e1.
Still, no search strategy is included and no inclusion/exclusion criteria are mentioned. Although you changed your topic considerably after my first review, the reader is interested also in the changed search strategy.
Following your advice, we have presented our search strategy in the section entitled “Introduction”:
“We conducted a comprehensive literature review using electronic databases such as Pubmed, Science Direct and Google Scholar, and took into account articles published in English between 1988 and April 2021. Keywords such as: “biomarker”, “endometriosis”, “glycoproteins”, “urocortin”, “immunological markers”, “oxidative stress”, “microRNA”, “lncRNA”, “urinary biomarkers”, and various combinations of the above were used. Publications were selected if they related to the studies investigating potential biomarkers detected in women with endometriosis. In addition, we manually reviewed the references for each article to find potentially missed studies. As a result of this, we identified a total of 2,073 relevant articles related to the topic of interest. Having considered the inclusion/exclusion criteria and eliminating duplicates, 93 studies were selected for analysis.
Inclusion criteria for the study selection:
- The samples of biomarkers could be collected from serum, plasma, whole blood, tissue, urine
- Randomized clinical trials, systemic reviews, meta-analyses
- Animal, human, in vitro studies.
Exclusion criteria:
- Case reports, conference summaries, comments
- Insufficient data
- Not accessible as a full-text article for review
- Language other than English
- Studies conducted on non-mammalian species.
The initial aim of the study was to identify potential novel diagnostic markers for the diagnosis of endometriosis, however, during the review process it was suggested that the subject should be changed and urinary biomarkers and glycoproteins such as Cancer Antigen 125 (CA-125) were included.”
In the paragraph about urine biomarkers you missed Tokushige et al. 2011 in Fertil Steril with keratin 19 and also the reference of Liu et al. 2015 in Cochrane could be included.
We agree that these publications are very important and valuable. We have presented the suggested reports on urine biomarkers as follows:
[…] “However, this area of interest requires further research investigating patients with a broader spectrum of endometriosis [80, 81].
[…] More in-depth evaluation of VDBP across the spectrum of endometriosis, especially in the luteal phase, is needed. An important aspect of further research is to determine the role, if any, that VDBP plays in endometriosis [81].
[…] However, Tokushige et al. [87] used proteomic techniques and thus demonstrated the presence of CK-19 in urine in women with endometriosis, comparing with a control group. In this survey, study group is not big enough to conclusively specify the value of CK-19 as biomarker for endometriosis. It is necessary to perform larger scale studies involving more patients. Presumably, CK 19 is not reliable as a diagnostic biomarker, as confirmed by Liu et al. [81].”
References:
- Liu, E.; Nisenblat, V.; Farquhar, C.; Fraser, I.; Bossuyt, P.M.; Johnson, N.; Hull, M.L. Urinary biomarkers for the non-invasive diagnosis of endometriosis. Cochrane Database Syst Rev. 2015, 12, CD012019.
- Tokushige, N.; Markham, R.; Crossett, B.; Ahn, S.B.; Nelaturi, V.L.; Khan, A.; Fraser, I.S. Discovery of a novel biomarker in the urine in women with endometriosis. Fertil Steril. 2011, 46-9.
In the manuscript by Florkowski 2008 (ref. no. 89), the author also mentioned that the likelihood ratio is more intelligible to users. Thus, I would like to know whether it is possible to include it also in the present review.
Thank you very much for your perceptive comments. We have expanded the issue of the likelihood ratio as follows:
"It should also be pointed out that LR seems to be a more intelligible way of conveying the accuracy of a diagnostic test to the user. Bearing in mind that application of LR enables a more appropriate interpretation of tests, it merits further adoption into clinical practice [Florkowski et al., 2008].”
Thank you very much for finding the time to read our manuscript and make helpful remarks. We would like to take this opportunity to express our gratitude for valuable and useful comments which have substantially contributed to improving the quality of our paper.
Yours faithfully,
Assoc. Prof. Zaneta Kimber-Trojnar, M.D., Ph.D.
Chair and Department of Obstetrics and Perinatology
Medical University of Lublin
Jaczewskiego 8, 20-090 Lublin, Poland
Tel: +48 81 7244 769;
Fax: +48 81 7244 841
E-mail: zkimber@poczta.onet.pl

This manuscript is a resubmission of an earlier submission. The following is a list of the peer review reports and author responses from that submission.
Round 1
Reviewer 1 Report
As indicated by the authors the review is focused on novel diagnostic markers for endometriosis. Overall, the paper is clearly written and interesting to read. It should be specifically highlighted that recent findings are discussed in the review. However, I need to mention that the study seems a bit descriptive, lacking authors’ own opinion. It would be helpful if the author provides advantage and disadvantage of diagnostic markers for endometriosis and put citation into table 1.
Reviewer 2 Report
Kimber-Trojnar et al. submitted a manuscript dealing with novel diagnostic markers for endometriosis. The manuscript is well written, however, several issues have to be answered.
Major Issues
- The definition of novel biomarkers remains unclear. It is not clearly stated whether there was a time restriction before March 2021. I also wonder why CA-125 was discussed because this is an old marker.
- The inclusion/exclusion criteria are missing, besides PRISM.
- A lot of references concerning several markers are missing. I will show only few examples. In the section about CA-125 O et al. 2019 and Vodolazkaia et al. 2012 are missing to mention some. In the section about sICAM-1 O et al. 2019, Kuessel et al. 2017 and Vodolazkaia et al. 2012 are missing to mention some oversights. In the section about microRNAs the work of Taylor is completely missing.
- I wonder why the authors included tissue biomarkers such as integrins. Besides that there are more tissue biomarkers. Furthermore the integrin section does not fit to the preceding parts of the manuscript.
Minor Issues
- It is still unclear whether inflammation is the cause or consequence or maybe both in the pathogenesis of endometriosis. This should be discussed.
Reviewer 3 Report
Dear Author,
I read with great interest the manuscript titled “Novel diagnostic markers for endometriosis”.
The Authors performed a review on diagnostic markers for the diagnosis of endometriosis, considering glycoproteins, growth factors and peptides, immunological markers, oxidative stress markers, microRNAs and integrins.
The Authors conclude that none of the marker considered meets the sensitivity and specificity required to be used as diagnostic marker itself, and further studies are required to evaluate a combination of them.
The study deals with a really interesting and relevant topic and the structure of the paper is well thought.
The manuscript is properly written with a concise introduction to the topic, the review of literature is not redundant, clear and relevant and the division into subsections makes reading more effective. References are adequate.
Furthermore, the study is written with a proper language.
This is a comprehensive literature review. If the intention of the authors was to perform a systematic review of the literature, I recommend to follow the PRISMA guidelines by reporting the flow chart of the selection of the papers considered and to describe the quality and the bias of the studies included according to specific tools.
Reviewer 4 Report
The review addresses an interesting question, which is regularly updated.
Importantly, the proposed title and aim are discordant: the proposed title indicates "novel diagnosis markers" whereas the stated aims are "to describe and discuss the current status of biomarkers of endometriosis in serum" (line 53). This is also discordant with IHC method (see below). After significant rewriting of the manuscript, I encourage the authors to resubmit an article addressing the potential of non invasive biomarkers for early diagnosis of asymptomatic patients with endometriosis, for instance. As this is a review, the authors should be more conclusive to demonstrate some original work (Could they stress which are the most interesting options for future studies?). Also, they do not take sufficient account the limitations: please, discuss the required skills (ie. learning curve for USG for accurate diagnosis of endometriosis), the time and cost of analysis, the technical aspects for miRNA/lncRNA profiling, the different type of samples and method of collection as a well known source of biais, ...
Major comments
MJ1: Please, for each method, consider the technical limitations, time and cost of analysis.
MJ2: The keyword "biomarker" must at least be included. Indeed, it identifies uncited references (ie. PMID: 32143439, 28189296 or 32019326) exploring the lncRNA, the auto antibodies, the urinary biomarkers, and numerous candidates not listed - such omission is striking for a review on the topic!
MJ3: Concerning integrins, please justify why discussing non soluble adhesion molecules as biomarkers (a fortiori for non invasive diagnosis). Whether the question is for diagnosis tool, other validated and more relevant markers are already used by anatomopathologists. Please note that dysregulation of integrin b1 expression is commonly reported and presumably not a good biomarker for endometriosis, in agreement with the author's conclusions. I suggest to purely delete this section and to adapt the conclusion and the table accordingly.
MJ4: The diverse types of endometriosis are considered distinct entities.Therefore, the authors should consider that the biomarkers will be different. This essential concept is totally absent in the manuscript in its current form.
Minor comment:
Mn1: Given that no candidate as emerged as a validated biomarker alone, it should be stressed further that a combination of positive and negative biomarkers should be considered. Please, consider adding a section (perspectives) on which biomarkers should be used and how.
Mn2: Looking at ROS, please use better references (ref 43 is poorly informative on ROS). The authors should mention the Cochrane statement (with meta-analysis) on these biomarkers (PMID 27132058). There is place for elaboration given the interest for ROS as biomarkers in the past (PMID 22791380).
Mn3: On figure 1, the uterus on the left is redundant and the right panel could be enlarged. Please identify the three types of endometriosis. Correct spelling is "inflammatory".